# Preserving privacy and video quality through remote physiological signal removal
Saksham Bhutani [1,4], Mohamed Elgendi [2,3,4,5] & Carlo Menon [1,5]

The revolutionary remote photoplethysmography (rPPG) technique has enabled intelligent devices to estimate physiological parameters with remarkable accuracy. However, the continuous and surreptitious recording of individuals by these devices and the collecting of sensitive health data without users' knowledge or consent raise serious privacy concerns. Here we explore frugal methods for modifying facial videos to conceal physiological signals while maintaining image quality. Eleven lightweight modification methods, including blurring operations, additive noises, and time-averaging techniques, were evaluated using five different rPPG techniques across four activities: rest, talking, head rotation, and gym. These rPPG methods require minimal computational resources, enabling real-time implementation on low-compute devices. Our results indicate that the time-averaging sliding frame method achieved the greatest balance between preserving the information within the frame and inducing a heart rate error, with an average error of 22 beats per minute (bpm). Further, the facial region of interest was found to be the most effective and to offer the best trade-off between bpm errors and information loss.

The ubiquitous presence of video-capturing devices in our daily lives is undeniable—from smartphones to personal computers and security cameras, these gadgets have become an inseparable part of our everyday routines, recording our movements and documenting our most private moments. However, recent advancements in computer vision have taken this technology to a whole new level, allowing intelligent devices to do much more than simply capture footage. Thanks to the revolutionary remote photoplethysmography (rPPG) technique, these devices can now estimate users' physiological parameters with remarkable accuracy[1]. Photoplethysmography (PPG) utilizes contact-based photoreceivers to measure the light reflected from certain regions of the skin, and it works by detecting light absorption variations, which are influenced by the amount of blood present under the skin[2,3]. PPG can be used to monitor a variety of health conditions, including heart rate (HR)[4], respiration rate[5], heart rate variability[6], and blood oxygen levels[7]. Meanwhile, rPPG can measure PPG signals remotely by detecting subtle changes in the color of a person's facial skin using consumer-grade digital cameras in ambient light[8].

While this breakthrough technology has the potential to transform the way we monitor our health, it also raises serious privacy concerns. The continuous and surreptitious recording of individuals by these devices and

the collecting of sensitive health data without users' knowledge or consent present a troubling prospect. For instance, this information is susceptible to misuse for manipulation in a variety of contexts, including marketing, negotiations, and other situations in which personal information can be weaponized against individuals. Thus, it is imperative to adopt techniques that can effectively conceal the rPPG signals captured from facial videos while simultaneously maintaining video quality, particularly to address the serious privacy concerns associated with the use of rPPG technology and to ensure video recordings remain useful for their intended purposes. Achieving this requires advanced technical methods that can accurately extract and remove PPG signals from facial videos without disturbing other visual features.

The current body of literature on modifying video frames using advanced techniques presents a range of promising options, yet most require estimation of the red-green-blue (RGB) time-series signal from a sequence of frames, limiting their practicality for real-time use. Furthermore, several of these methods require high computational power, making them impractical for widespread adoption. To address these issues, our study analyzes several frugal alternative video-modification techniques that require minimal computational resources and that can run directly on

[1]Biomedical and Mobile Health Technology Research Lab, ETH Zürich, Zürich, Switzerland. [2]Department of Biomedical Engineering and Biotechnology, Khalifa University of Science and Technology, Abu Dhabi, UAE. [3]Healthcare Engineering Innovation Group (HEIG), Khalifa University of Science and Technology, Abu Dhabi, UAE. [4]These authors contributed equally: Saksham Bhutani, Mohamed Elgendi. [5]These authors jointly supervised this work: Mohamed Elgendi, Carlo Menon. ✉e-mail: mohamed.elgendi@ku.ac.ae; carlo.menon@hest.ethz.ch

hardware with limited compute resources. By exploring these techniques, we hope to provide a more accessible and practical approach to video modification that can benefit a wider range of users and applications.

## Methods

### Prior work

PPG is typically recorded using a device that sends light into the skin and measures the amount reflected. The device may use a laser, light-emitting diode, or other light source, and it is typically placed on the fingertip, wrist, or earlobe. The device can then detect changes in blood volume in the skin by measuring the amount of light absorbed or reflected[2,3]. This information can be used to calculate various physiological parameters, including HR[4], respiration rate[5], heart rate variability[6], and blood oxygen levels[7], and they can be further used to determine various cardiovascular diseases[9].

**rPPG estimation.** The rPPG technique is non-invasive, employing a video camera to measure fluctuations in blood volume in the skin. To estimate rPPG, various signal processing and machine learning approaches have been proposed. A typical signal processing pipeline, as proposed by Boccignone et al.[10], for estimating HR using rPPG is described below.

Figure 1 provides a visual representation of the rPPG estimation pipeline. Initially, the subject's face is extracted from the video in the face extraction step, after which region of interest (ROI) processing is performed on the extracted face. The literature has analyzed many different ROIs, including the entire face without the eyes and lips or patches from specific facial regions, including the forehead and cheek[11,12], but Fig. 1(b) illustrates the selection of the entire face as the ROI. The ROIs are then utilized to calculate average RGB intensities over time, as shown in Fig. 1(c). The calculated RGB time series is then divided into multiple windows that can be subjected to signal preprocessing, such as detrending, frequency-selective filtering, or normalization (Fig. 1[d]).

Next, blood volume pulse (BVP) is estimated from filtered RGB signals (Fig. 1e), for which various methods have been proposed in the literature, including:

- GREEN[8]: This method focuses on the green channel of video frames to estimate the rPPG signal, as it is believed to contain the most information about blood volume changes in the skin. It filters other channels to concentrate on the green channel.
- Independent Component Analysis (ICA)[13]: This algorithm uses ICA to decompose filtered RGB signals to recover three source signals. The second component of ICA is used as it contains a considerable amount of the BVP signal.
- Chrominance (CHROM)[14]: This algorithm estimates the rPPG signal by performing color channel normalization, which assumes the ratio of two color channels is not influenced by the motion of light.
- Plant-Orthogonal-to-Skin (POS)[15]: This algorithm uses the POS tone in the normalized RGB space.
- Local Group Invariance (LGI)[16]: This algorithm provides features invariant to action and motion based on differentiable local transformations.

Finally, beats per minute (bpm) estimation is performed by analyzing the spectral content of the BVP signal using power spectral density (Fig. 1f) or the short-time Fourier transform. The bpm values, estimated from each window, are then combined to construct the bpm time series. Across multiple studies[4,10], the LGI and POS algorithms have been proven the best methods for BVP estimation.

Various machine learning-based methods for rPPG estimation have also been discussed in the literature, one of which is the convolutional neural network to estimate heart rate (HR-CNN)[17] where the convolutional neural network is divided into an extractor and an HR estimator. The extractor estimates a scalar output for a frame, which is then run over a sequence of frames to produce a sequence of scalars. The HR estimator then uses this sequence to extract the HR. Another method called the two-dimensional convolutional attention network (2D-CAN)[18], involves two parallel models: appearance and motion. Using multiple convolutional layers, the appearance model takes a frame and extracts an attention map, which helps determine the region of skin containing physiological signals, similar to the ROI processing step in the signal processing methods. Conversely, the motion model takes the normalized difference between two consecutive

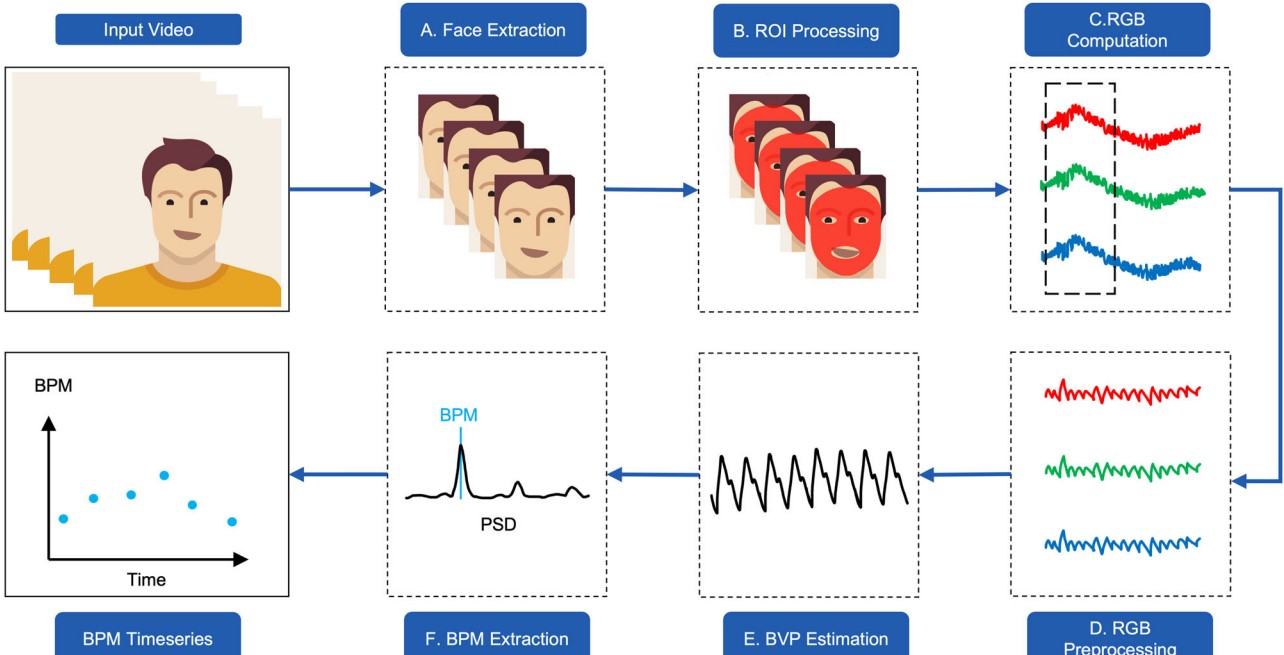

**Fig. 1 | Overview of the remote photoplethysmography (rPPG) estimation process.** This figure illustrates the workflow of rPPG-based heart rate estimation, from video input to beats per minute (BPM) output. The process includes region of interest (ROI) processing, red, green, blue (RGB) computation, RGB preprocessing, blood volume pulse (BVP) signal estimation, and heart rate extraction using power spectral density (PSD) analysis.

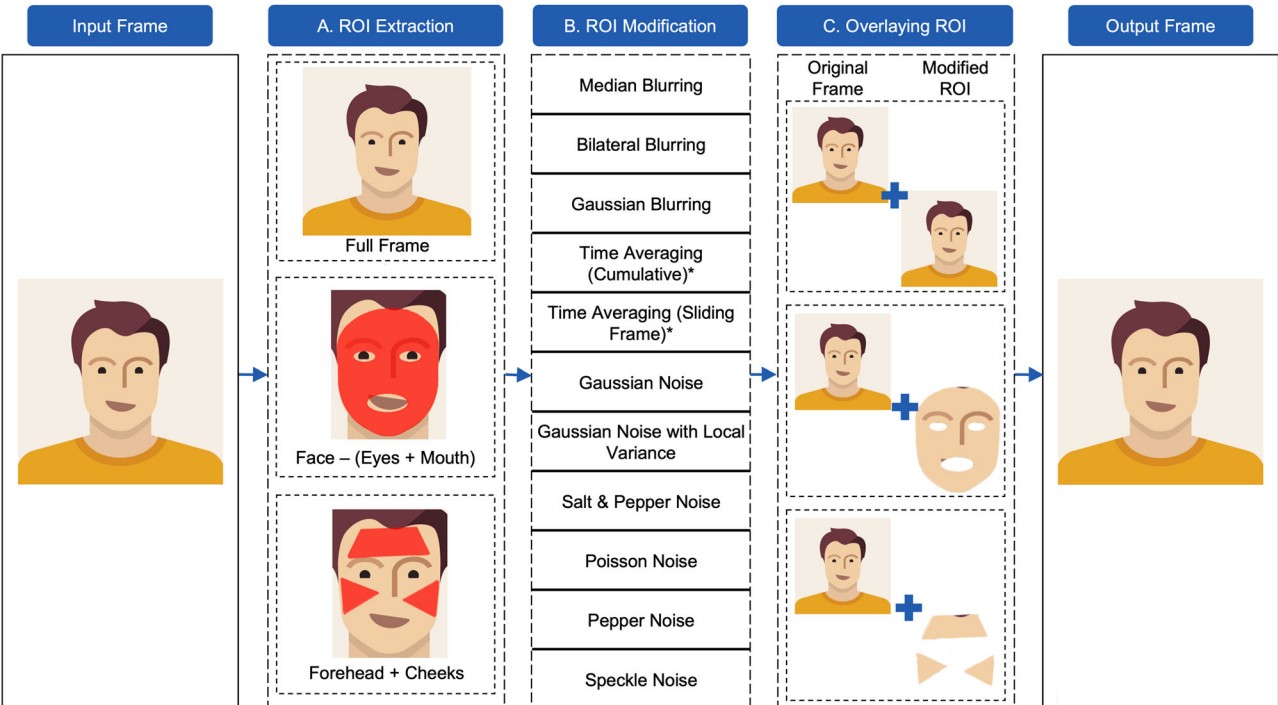

**Fig. 2 | Overview of the frame modification pipeline.** This figure outlines the three-step process of frame modification: region of interest (ROI) extraction, ROI modification, and ROI combination. The pipeline illustrates the application of 11 different modification techniques to three distinct ROIs: full frame, face excluding eyes and mouth, and forehead and cheeks only. The process begins with ROI extraction from a frame, followed by the application of modification techniques, and concludes with the modified ROI being overlaid back onto the original frame. Techniques requiring multiple frames for modification are denoted with asterisks.

frames and extracts a feature map using 2D convolutional layers. This model prioritizes only the features that the appearance model deems necessary by multiplying the feature map at two stages by the attention map provided by the appearance model. Finally, the motion model gives a scalar value, and the sequence of scalar values for multiple frames produces the BVP time-series signal. Although machine-learning models like HR-CNN and 2D-CAN have demonstrated promising performance in rPPG estimation, they are unable to operate in real time with limited computational resources.

**rPPG elimination techniques.** In the current literature, several techniques exist to remove or spoof the rPPG signal when modifying facial videos, one of which, proposed by Chen and McDuff[19], is called motion elimination, where each video frame is divided into $8 \times 8$-pixel blocks and then an L-level Laplacian pyramid is computed to estimate a time series. They also used principal component analysis to select components of interest based on the power spectral density, multiplied the selected component with a constant scalar A, and reconstructed the image using these components to eliminate the rPPG signal. However, this approach considers only the steady case and not voluntary motions, such as rotation and translation, made by the subject. Another method, called PulseEdit[20] edits the PPG signal by adding perturbation frames to the original video, which are created by estimating the RGB time series from multiple ROIs, detrending these signals, and using Pearson's correlation coefficient to maximize similarities between the time series and the target PPG signal. Another technique proposed by Li et al.[21] involves creating injection templates by extracting the rPPG signal from the video using the face as the ROI and adding sine waves to the signal. The templates are fuzzed using convolution kernels and superimposing them back onto the original video frames. Finally, Privacy-Phys[22], which uses a pre-trained three-dimensional convolutional neural network model, was conditioned to modify the video while minimizing information loss.

These methods are all based on video processing and cannot be used on individual frames. Moreover, due to their high computational costs and limited real-time applications, most existing approaches are incompatible with edge platforms, such as smartphones and security cameras. To address this limitation, we have explored and developed lightweight image modification methods that can be run efficiently on edge platforms with little to no inter-frame dependence.

**Frame modification pipeline**
In this study, a frame modification pipeline was developed to increase privacy by hiding the rPPG signal in facial videos, a graphical representation of which is shown in Fig. 2. The pipeline consists of three main steps: ROI extraction, ROI modification, and ROI combination. The first aim of the pipeline is to reduce information loss during frame modification, as previous research[11,12] demonstrated that specific facial regions are more influenced by blood volume changes. Modifying these specific areas would be as effective as modifying the entire frame while minimizing information loss. However, extracting these regions is computationally expensive. Thus, to ensure a balance between information loss and the modification speed, we tested three different ROIs.

The first ROI is the full frame, where everything included within the frame is considered the ROI. The second ROI is the facial skin, where everything but the eyes and mouth is considered the ROI, and this is achieved by detecting the facial mesh using MediaPipe[23], creating a convex hull using the outermost points of the mesh, and removing the eyes and mouth. Finally, we used the forehead and cheeks as the third ROI, which were extracted similarly by creating three separate convex hulls from the facial mesh points for the forehead and the left and right cheeks.

To modify the ROIs, we tested 11 different frame modification methods, the effects of which are summarized in Table 1. To achieve this, we employed the built-in functions of Open Computer Vision[24] for blurring operations, one of which is median blurring (MB), which calculates the median value of each pixel in the ROI based on the

**Table 1 | Summary of all techniques used in this study for frame modification**

| | Modification method | Summary |
|---|---|---|
| Blurring (B) | Median (MB) | MB is a non-linear smoothing technique that replaces each pixel's value with the median value of its neighboring pixels, effectively reducing variations in RGB values in the region of interest (ROI) while preserving edges and fine details[28]. |
| | Gaussian (GB) | GB is a linear smoothing technique that convolves the ROI with a Gaussian kernel, reducing high-frequency components and resulting in a smoother ROI with reduced noise[29]. |
| | Bilateral (BB) | BB is a non-linear smoothing technique that preserves edges and fine details while reducing variations in RGB values. It uses a weighted average of neighboring pixels, where the weights depend on both spatial proximity and pixel intensity differences[30]. |
| Time-Averaging (TA) | Cumulative (TA-C) | TA-C averages the pixel values of all frames in the ROI, creating a temporal smoothing effect. |
| | Sliding Frame (TA-S) | TA-S averages the pixel values of the last f frames in the ROI, creating a temporary temporal smoothing effect. |
| Noise (N) | Additive Gaussian (AGN) | AGN adds random values to each pixel in the ROI, following a Gaussian distribution. This type of noise can be modeled and removed using statistical methods[29]. |
| | Additive Gaussian with local variance (AGN-L) | AGN-L adds random values to each pixel in the ROI, following a Gaussian distribution, but with a variance that depends on the local image content[29]. |
| | Salt and pepper (SPN) | SPN is an impulse noise that randomly sets some pixels in an image to their minimum or maximum intensity value[31]. |
| | Poisson (PoN) | PoN is a type of noise that arises in low-light imaging situations due to the random nature of photon arrivals. This noise is modeled using the Poisson distribution[31]. |
| | Pepper (PeN) | In PeN, pixels are randomly replaced with minimum intensity values[31]. |
| | Speckle (SN) | SN is created due to the interference of scattered waves, which results in constructive and destructive patterns, leading to the appearance of bright and dark spots in the image, known as speckles[31]. |

surrounding pixels. The aperture of the algorithm, specified by the kernel size, determines which pixels are considered. Another technique we used was the Gaussian filter, which convolves the frame with a Gaussian kernel, the values for which are sampled from a 2D Gaussian function, where the standard deviation of space ($\sigma_s$) is used to calculate the kernel size. In this study, we used a default kernel size of 5 for both MB and GB. The bilateral filter is another technique that uses the Gaussian filter, along with two additional terms: a normalization factor and another Gaussian term, known as the range weight with $\sigma_r$. This filter ensures that only pixels with intensity values similar to that of the central pixel are considered for blurring, while sharp changes in intensity are preserved. In this study, we used a default value of 75 for both $\sigma_s$ and $\sigma_r$.

The time-averaging operations differ from others, as they have minimal inter-frame dependence, taking an average of the pixels in the ROI over time, which minimizes the variations in RGB channels and reduces the visibility of BVP changes. The cumulative time-averaging (TA-C) technique involves averaging each pixel in the ROI from the beginning frame ($t = 0$) to the current frame ($t$). Conversely, the time-averaging sliding window (TA-S) technique only averages the pixels in the ROI in the preceding $f$ frames. We used a 20-frame sliding window in this study. The Scikit-Image[25] python library was utilized in our study to add noise, multiple types of which were incorporated, including additive Gaussian-distributed noise (AGN), which follows a normal distribution and is added to a signal to replicate real-world noise, and additive Gaussian-distributed noise with local variance (AGN-L), which involves adding random noise with a local variance to the ROI following a Gaussian distribution. The local variance may differ across various parts of the ROI, resulting in a more realistic and detailed representation of noise. For Poisson noise (PoN), pixels in the ROI are multiplied by a random number generated from a Poisson distribution whose mean corresponds to the intensity value of the pixel, while pepper noise (PeN) replaces random pixels in the ROI with 0 and salt and pepper noise (SPN) with either 0 or 1. Finally, speckle noise (SN) is generated by multiplying the pixels with random noise that has a multiplicative nature. All noise types were used with default parameters.

In the third step of the modification pipeline, we overlay the edited ROI over the original frame. The goal of this pipeline was to modify facial videos such that the rPPG signal was hidden or distorted, while still maintaining the

facial appearance. However, the choice of modification method depends on the specific requirements of the application.

**rPPG estimation**. To analyze the impact of the frame modification pipeline, a standard rPPG estimation pipeline, similar to that shown in Fig. 1, was created using the pyVHR framework[10]. The facial skin was selected as the ROI using the default skin extractor, and RGB signals were extracted and averaged from 100 equispaced landmarks on the forehead, cheeks, and nose. The RGB time series and the ground truth fingertip PPG signals were divided into overlapping windows of 8 seconds, and a sixth-order Butterworth bandpass filter ranging from 0.65 to 4.0 Hz was applied for preprocessing. Five BVP extraction methods were tested, including GREEN[8], ICA[13], CHROM[14], POS[15], and LGI[16], but deep learning-based techniques were not used due to their slow nature and high computational requirements.

**Dataset**. To test the performance of the algorithms in this study, we used the LGI-PPGI dataset, collected by Pilz et al.[16], consisting of facial videos recorded from 25 subjects (20 males and 5 females), as well as a ground truth PPG signal recorded from a CMS50E pulse oximeter. The videos are recorded at 25 frames per second (fps), and the corresponding PPG data are recorded at 60 Hz. Of note, only data for six subjects (5 males and 1 female) were publicly available at the time of writing and thus used in the analysis, which involved four different scenarios, as follows:

1. Resting: The subjects are sitting still indoors, avoiding any head motion.
2. Rotation: The subjects are asked to make arbitrary head movements.
3. Talk: The subjects are having a conversation outdoors in an urban environment.
4. Gym: The subjects are engaging in an indoor workout using a bicycle ergometer.

This dataset contains a wide range of recording conditions, including different lighting, subject motions, and facial expressions, offering a realistic and challenging testbed with which to evaluate algorithm performance. In addition, using a dataset with multiple subjects in varying scenarios allows the results to be generalized to a wider population. Further, this dataset has

been widely used in the literature to evaluate rPPG algorithms, contributing well-established benchmarks for comparison.

**Evaluation metrics.** To evaluate the algorithm's performance of hiding the rPPG signal in the modification of facial videos, we used several evaluation metrics, which were calculated based on a comparison between the ground truth fingertip PPG signal, provided by the LGI-PPGI, and the new rPPG signal, estimated from the modified video. The evaluation metrics used in this study were as follows:

(i) Beats-per-minute difference To quantify the degree of modification in the rPPG signal, we adopted HR as the physiological parameter of interest, typically defined in bpm and able to be estimated from the BVP signal through identification of the peak of the power spectrum. To evaluate the accuracy of our rPPG estimation pipeline, we calculated the average absolute difference between the HR values extracted from the ground truth PPG signal and the estimated rPPG signal, a metric that quantifies deviations between the peak frequency of the rPPG signal and the PPG signal in the power spectrum and that can be defined as follows:

$$|\Delta\text{bpm}| = \frac{1}{n}\sum_{i=1}^{n}|h_i - \hat{h}_i| \qquad (1)$$

where $h_i$ is the beats extracted from the ground truth PPG signal per minute, $\hat{h}_i$ is the bpm from the estimated rPPG signal, and n is the number of samples. $|\Delta\text{bpm}|$ is a robust metric that is resistant to outliers and is useful for evaluating the overall accuracy of the HR estimation.

(ii) Mean squared error (MSE) MSE was adopted to quantify the data loss between corresponding frames, as it calculates the average squared difference between the pixel values of the modified and original frames, providing a quantitative measure of the amount of information lost during modification. The MSE is calculated using the following formula:

$$\text{MSE} = \frac{1}{mn}\sum_{i=1}^{m}\sum_{j=1}^{n}(I_{i,j} - K_{i,j})^2 \qquad (2)$$

where $I_{i,j}$ is the pixel value of the $(i, j)$-th pixel in the original frame, $K_{i,j}$ is the pixel value of the corresponding pixel in the modified frame, and $m$ and $n$ are the dimensions (height and width) of the frames, respectively.

(iii) Overall evaluation score A metric, called the overall evaluation score (OS), was developed to assess the complete performance of the modification methods. It is a combination of other metrics and is calculated using the following formula:

$$\text{OS} = \frac{1}{2}(|\Delta\text{bpm}_n| + (1 - \text{MSE}_n)) \qquad (3)$$

The first part of the formula, $|\Delta\text{bpm}_n|$, represents the absolute difference between the bpm values extracted from the ground truth and the modified rPPG signals, and it is normalized using the min-max normalization technique. The second part of the formula, called the data retention factor $(1 - \text{MSE}_n)$, represents the amount of data retained between the modified and original frames, and it is also normalized using the min-max normalization technique. By combining these two normalized metrics, the OS provides a balanced assessment of the modification methods, considering both the accuracy of the bpm values and the amount of data loss. An ideal modification method should maximize bpm errors while minimizing data loss.

(iv) Frames per second (fps) Fps is a metric used to quantify the number of frames displayed each second in a video, and it measures the video's performance in real-time scenarios, where a higher fps value indicates the video can be modified in real time without a lag or delay in the output. The fps value can be computed as the reciprocal of the time taken to process a single frame of a video, which is denoted by $T_{frame}$, but it is important to note that fps is dependent on the hardware used and may vary with different hardware configurations.

$$\text{fps} = \frac{1}{T_{frame}} \qquad (4)$$

These evaluation metrics were selected based on their ability to measure the accuracy of the estimated rPPG signal, the loss in video quality, and the algorithm's overall performance. They provide a comprehensive evaluation of the algorithm and allow fair comparisons with other methods.

## Statistics and reproducibility

This study does not employ traditional statistical tests (e.g., t-test, chi-square test, ANOVA, Pearson correlation) and instead evaluates the modification methods using alternative evaluation metrics that are more relevant to our objectives, namely: $|\Delta\text{bpm}|$, MSE, overall score, and fps. The source code[26] employed in this study has been made publicly accessible to promote reproducibility and extendibility. Specifically, the pyRemoval package within the source code encompasses all the tools necessary for video processing, including the modification methods discussed in this study. Additionally, the package offers the functionality to analyse the processed videos, including the assessment of information loss and runtime speed. Importantly, the source code is not restricted to the LGI-PPGI dataset and is adaptable to other datasets. Thorough installation instructions are provided alongside the source code, ensuring straightforward deployment. Extensive testing has been conducted across different platforms to verify the compatibility. A detailed demonstration notebook is included, with comprehensive guidance on the utilization of the package. Our aim is to establish this codebase as a foundational resource for researchers to build upon. To facilitate this, we have provided boilerplate code along with detailed instructions for creating filters for video modification, integrating different ROIs, and analyzing methods using alternative metrics to quantify information loss and runtime speed.

## Reporting summary

Further information on research design is available in the Nature Portfolio Reporting Summary linked to this article.

## Results

Table 2 presents a comprehensive overview of the results for each frame modification method and rPPG estimation technique, averaged across all subjects and activities. The ROI used for the evaluation was the face region without the eyes and mouth. Meanwhile, the OS was estimated by min-max normalization $|\Delta\text{bpm}|$ for all rPPG techniques and by normalizing the MSE. All values have been rounded to two decimal places for better readability.

### Error in beats per minute estimation

A box plot shown in Fig. 3 was employed to determine the modification method that causes the maximum error in bpm estimation, the results of which revealed that the TA-C method produced the highest error in bpm, followed by the TA-S, SPN, and PeN methods. It is worth noting that MB, Gaussian blurring (GB), and bilateral blurring (BB) exhibited similar errors to the unedited videos. The suboptimal bpm estimation in the unedited videos is possibly due to the presence of various random movements of the subjects during gym and rotation activities. It was also observed that all the outliers were from the gym activity videos. The maximum error of 52 bpm was observed in the subject named CPI during the gym activity, while the minimum error of 1.3 bpm was observed in the subject named Angelo during the resting activity. We also computed a metric to identify the rPPG technique that could best withstand frame modification, which was $1 - |\Delta\text{bpm}_n|$, as it achieved a score ranging between 0 and 1. Our analysis revealed that the LGI, CHROM, POS, and GREEN were resilient, all having

**Table 2 | Evaluation metric results for all modification methods for each rPPG estimation technique**

| Modification Method | |Δbpm| | | | | | MSE | OS |
|---|---|---|---|---|---|---|---|
| | CHROM | POS | LGI | GREEN | ICA | | |
| No Editing (NE) | 8.84 | 5.09 | 8.19 | 16.30 | 16.71 | 0.00 | 0.51 |
| Median Blur (MB) | 8.85 | 5.47 | 8.22 | 15.81 | 16.74 | 52.74 | 0.42 |
| Gaussian Blur (GB) | 8.89 | 5.24 | 8.10 | 15.68 | 17.78 | 50.68 | 0.44 |
| Bilateral Blur (BB) | 8.94 | 5.32 | 7.98 | 15.96 | 16.76 | 49.76 | 0.43 |
| Time-Averaging (TA-C) | 26.12 | 27.98 | 27.23 | 26.96 | 25.82 | 329.15 | 0.50 |
| Time-Averaging (TA-S) | 21.69 | 19.67 | 19.98 | 24.08 | 24.66 | 136.32 | 0.66 |
| Gaussian Noise (AGN) | 11.51 | 9.93 | 10.81 | 16.28 | 23.78 | 194.06 | 0.34 |
| Gaussian Noise (AGN-L) | 11.24 | 9.19 | 9.52 | 18.17 | 21.73 | 194.06 | 0.32 |
| Salt & Pepper Noise (SPN) | 14.13 | 11.46 | 12.92 | 19.23 | 25.64 | 265.21 | 0.31 |
| Poisson Noise (PoN) | 9.17 | 5.62 | 8.33 | 15.97 | 19.19 | 74.27 | 0.42 |
| Pepper Noise (PeN) | 11.71 | 10.45 | 12.17 | 17.74 | 23.68 | 254.13 | 0.27 |
| Speckle Noise (SN) | 9.78 | 6.21 | 8.56 | 15.74 | 18.63 | 84.40 | 0.41 |

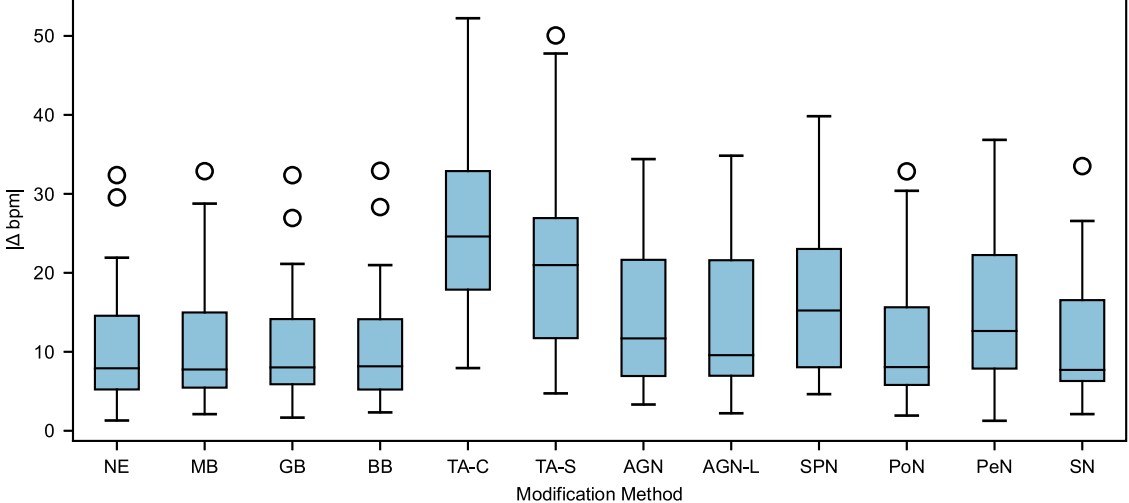

**Fig. 3 | Impact of frame modification methods in heart rate estimation accuracy.** This figure presents the average absolute difference in beats per minute (|Δbpm|) for various frame modification methods across different rPPG estimation techniques. The box and whisker plots represent the data distribution, where the median is shown by the horizontal black line, the colored boxes indicate the first and third quartiles and the whiskers extend from the box to the farthest data point within 1.5 times the inter-quartile range from the box. Outlier points are those lying past the end of the whiskers. The facial region excluding eyes and mouth was used as the region of interest (ROI). The results are estimated for each subject ($n = 6$) and each the activity ($n = 4$) in the LGI-PPGI dataset, averaged for all rPPG estimation techniques ($n = 5$) and over the entire length of the videos. All statistical values (means and quartiles) were computed using Python's matplotlib package. Source data provided in Supplementary Table 3.

score close to 0.8. Conversely, the ICA method was the least resilient with a score of 0.54.

**Information loss**
The information loss between the original and modified frames was measured using MSE, as illustrated in Fig. 4(a), the results of which indicate that the TA-C method generated the highest MSE score, followed by the SPN and PeN methods. In contrast, all three blurring methods produced comparable MSE values, around 50, while the MSE remained unchanged for AGN, regardless of whether any local variance was added.

**Runtime speed**
To assess the practical utility of the modification methods, we also computed the runtime speed in fps, shown in Fig. 4(b). The GB and MB methods exhibited the fastest fps values of 15, followed by the TA methods, with an fps of 11, while the PoN method was found to be

the slowest, with an fps of 2.5. It is worth noting that the TA-S approach will operate with a delay equivalent to the number of frames within the window and that the fps can be influenced by numerous factors, including the selected ROI and kernel size, which are discussed later in this section.

**Overall evaluation score**
Figure 5 displays detailed OS values for each rPPG technique, along with the average of all rPPG techniques. Among the modification methods, the TA-S technique consistently achieved the highest OS for all rPPG techniques, followed by the TA-C, GB, BB, and PoN methods, while the other noise addition methods achieved the lowest OS values. Notably, for the TA-C method, the OS is determined entirely by the absolute error in bpm, but when there is no editing, the score is mostly determined by the data retention factor due to the zero MSE. Concerning rPPG methods, ICA consistently achieved the highest OS due to poor HR estimation.

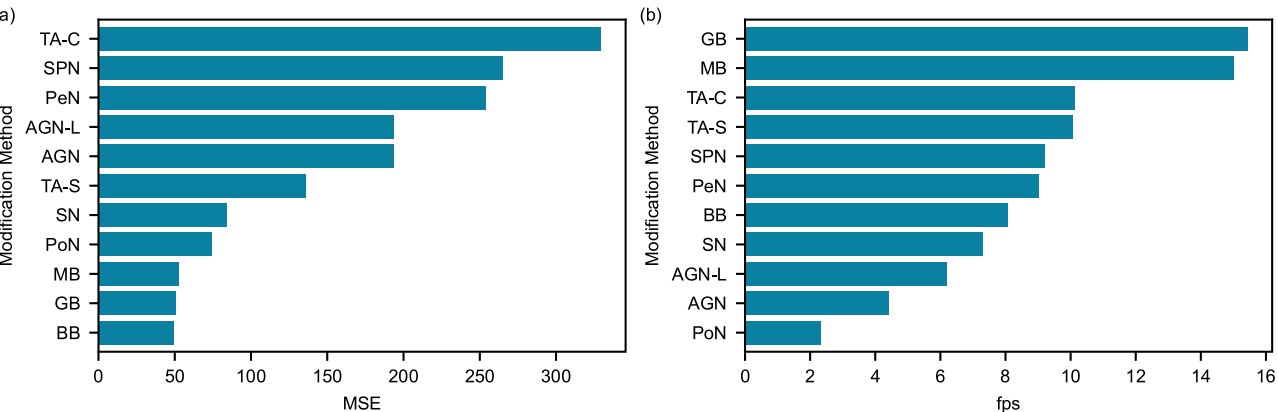

**Fig. 4 | Performance evaluation of frame modification methods. a** Mean squared error (MSE) quantifying data loss between original and modified frames. **b** Processing speed in frames per second (fps) for each modification method. Both metrics were evaluated using the facial region excluding mouth and eyes as the region of interest (ROI). The MSE value was computed by averaging for all activities ($n = 4$) and subjects ($n = 6$), while fps was estimated by running an analysis on a MacBook Pro with Intel i5 and was averaged over 1000 frames. Source data for (**b**) provided in Supplementary Table 4.

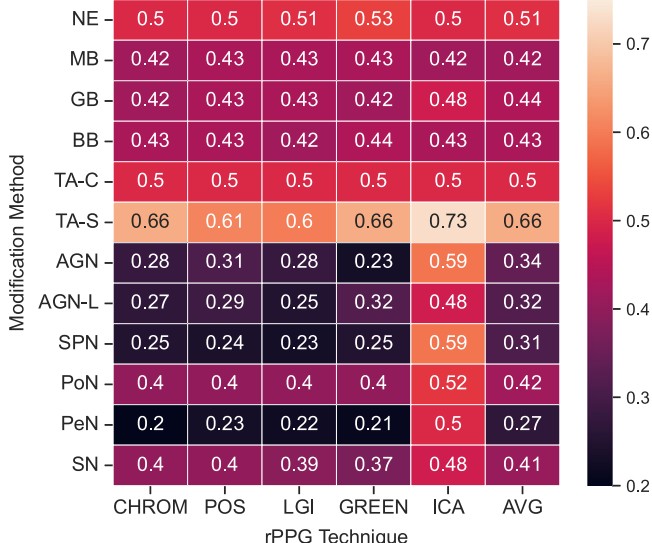

**Fig. 5 | Heatmap of overall evaluation scores for frame modification methods and rPPG estimation techniques.** This figure presents the overall evaluation score (OS) for each combination of modification method and remote photoplethysmography (rPPG) estimation technique. OS combines the average absolute difference in beats per minute and the mean squared error between frames to assess the complete performance of modification methods. AVG represents the average across all rPPG estimation techniques ($n = 5$). The color bar ranges from 0.2 to 0.75, with higher scores (lighter colors) indicating better performance. The results are averaged for all the subjects ($n = 6$) in the LGI-PPGI dataset and for all the activities ($n = 4$) over the entire length of the videos.

## Effect of Activities

The evaluation metrics for the study are presented in Table 3, which classifies the results based on four activities, namely resting, rotation, talk, and gym, considering all subjects and rPPG estimation techniques. It is observed that the resting activity generated the minimal number of bpm errors, followed by rotation, talking, and gym, and a similar trend was observed with the MSE. The OS was determined by min-max normalizing the bpm errors and the frame-to-frame MSE for all modification methods. The highest OS of 0.75 was obtained by the TA-S method in the gym activity. The OS for each activity and modification method is depicted in Fig. 6, which indicates the contributions of Δbpm and MSE to the OS.

In the resting activity, the OS was primarily determined by the $(1 − MSE_n)/2$ factor, while the contribution of $|\Delta bpm_n|/2$ was minimal. As the activities became more strenuous due to increased arbitrary head motions, the contribution of $|\Delta bpm_n|/2$ increased, resulting in an increased OS. On average, the OS increased in the order of resting>rotation>talk>gym. It is interesting to note that for the resting activity, the TA-C method showed the best performance, at par with the TA-S method, but it was one of the worst performers in the remaining activities, potentially because few artifacts are visible when all frames are averaged over time, particularly if there is minimal head movement.

## Influence of region of interest

In this investigation, we examined three distinct ROIs, as shown in Fig. 7, where the average bpm error was observed to be remarkably consistent across all three. The full-frame ROI had the highest average MSE, at approximately 600, while the facial ROI (excluding the mouth and cheeks) and the forehead with cheeks ROI displayed similar performances, with the OS indicating that the facial ROI was the best performer. Concerning fps, the full-frame ROI was the quickest, as it did not require facial detection or landmarking, while the forehead with cheeks ROI performed slightly worse than the facial ROI because it utilized three distinct convex hulls. It is important to note that in this study, we solely utilized rPPG-based physiological parameter extraction, which operates based on RGB intensity changes in the facial region. Nevertheless, research has shown that physiological signals can also be estimated using slight head movements caused by the ballisticardiogram[27]. The selective ROIs would not function with this approach, but the application of ballisticardiogram techniques is already limited because they do not work with voluntary head movements.

Additionally, a detailed examination of the various parameters pertaining to the modification methods was conducted, evaluating their impact on the overall performance. Supplementary Table 1 presents the influence of different kernel sizes on the GB and MB methods as table, while Supplementary Table 2 offers another table showcasing the effect of different window lengths on the TA-S method.

## Discussion

The TA-C method performed poorly due to the reduction in frame quality that it induced, resulting in a lower OS. Conversely, the TA-S method produced a slightly lower average bpm error, but with a considerably lesser reduction in frame quality, resulting in the highest OS. Several noise addition methods, such as SPN and AGN, were found to produce good OS values, whereas blurring methods consistently produced the lowest OS values, possibly due to the similarity between blurring methods and the rPPG pipeline's pixel values, as averaged from various regions. In terms of rPPG techniques, LGI was found to be the most resilient, producing the least amount of bpm error from modified frames, whereas ICA was found to be

## Table 3 | Evaluation metric results for all modification methods, divided into four activities

| Metric | |Δbpm| | | | | MSE | | | | OS | | | |
|---|---|---|---|---|---|---|---|---|---|---|---|---|
| Technique/Activity | Resting | Rotation | Talk | Gym | Resting | Rotation | Talk | Gym | Resting | Rotation | Talk | Gym |
| Median Blur (MB) | 3.78 | 7.36 | 14.14 | 22.68 | 25.10 | 46.43 | 49.77 | 108.12 | 0.50 | 0.52 | 0.59 | 0.62 |
| Gaussian Blur (GB) | 4.30 | 7.79 | 14.03 | 22.07 | 22.62 | 43.99 | 47.94 | 106.93 | 0.51 | 0.52 | 0.59 | 0.62 |
| Bilateral Blur (BB) | 3.67 | 7.60 | 13.94 | 22.64 | 21.70 | 43.35 | 46.32 | 106.60 | 0.50 | 0.52 | 0.59 | 0.63 |
| Time-Averaging (TA-C) | 20.13 | 21.64 | 25.09 | 47.23 | 76.84 | 328.87 | 483.58 | 476.37 | 0.63 | 0.37 | 0.25 | 0.51 |
| Time-Averaging (TA-S) | 12.34 | 15.90 | 22.78 | 44.56 | 24.52 | 150.17 | 174.71 | 225.66 | 0.60 | 0.50 | 0.55 | 0.75 |
| Gaussian Noise (AGN) | 5.52 | 9.95 | 21.98 | 23.37 | 141.89 | 167.63 | 200.16 | 302.80 | 0.39 | 0.41 | 0.52 | 0.42 |
| Gaussian Noise (AGN-L) | 5.02 | 9.44 | 20.58 | 24.26 | 141.90 | 167.63 | 200.16 | 302.79 | 0.39 | 0.41 | 0.50 | 0.43 |
| Salt & Pepper Noise (SPN) | 6.88 | 11.92 | 24.66 | 26.54 | 201.84 | 230.30 | 288.31 | 378.01 | 0.34 | 0.37 | 0.45 | 0.38 |
| Poisson Noise (PoN) | 4.28 | 8.15 | 14.95 | 23.04 | 42.95 | 66.12 | 70.11 | 139.72 | 0.48 | 0.50 | 0.58 | 0.59 |
| Pepper Noise (PeN) | 6.04 | 10.71 | 22.03 | 25.15 | 222.09 | 256.60 | 234.38 | 328.10 | 0.31 | 0.33 | 0.48 | 0.41 |
| Speckle Noise (SN) | 4.44 | 7.95 | 16.09 | 22.10 | 54.85 | 78.92 | 78.00 | 146.56 | 0.47 | 0.49 | 0.58 | 0.58 |

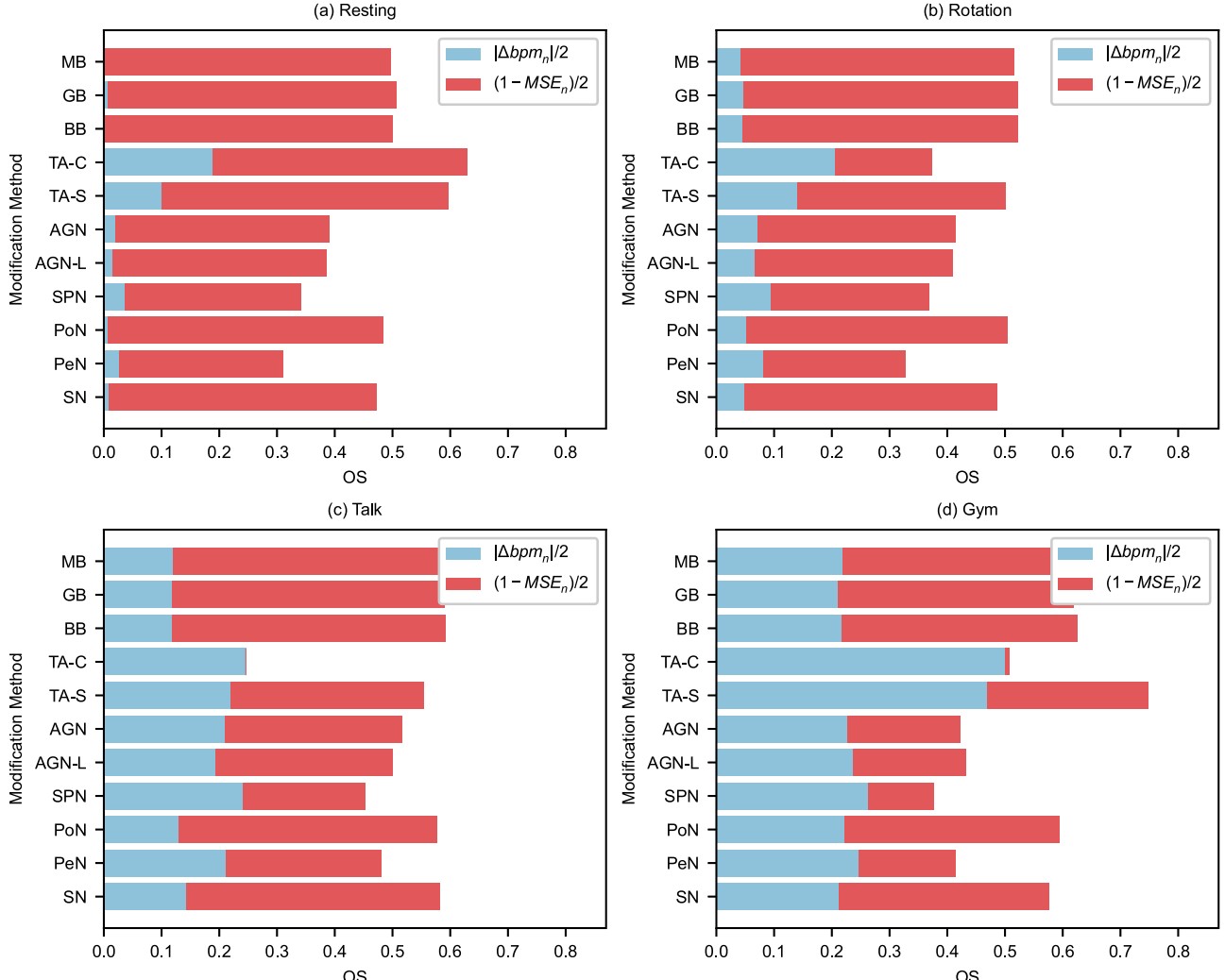

**Fig. 6 | Overall evaluation scores of video modification methods across different activities.** This figure presents the overall evaluation score (OS) for each video modification method, divided by activity: (**a**) resting, (**b**) rotation, (**c**) talk, and (**d**) gym. The OS is represented by the full length of the bar, while the blue portion represents the contribution of the normalized error in beats per minute ($|\Delta \text{bpm}_n|$) to the OS and the red portion the contribution of the normalized data retention factor $(1 - MSE_n)$ to the OS. The results are averaged for all the subjects ($n = 6$) in the LGI-PPGI dataset and for all the rPPG estimation techniques (n=5) over the entire length of the videos.

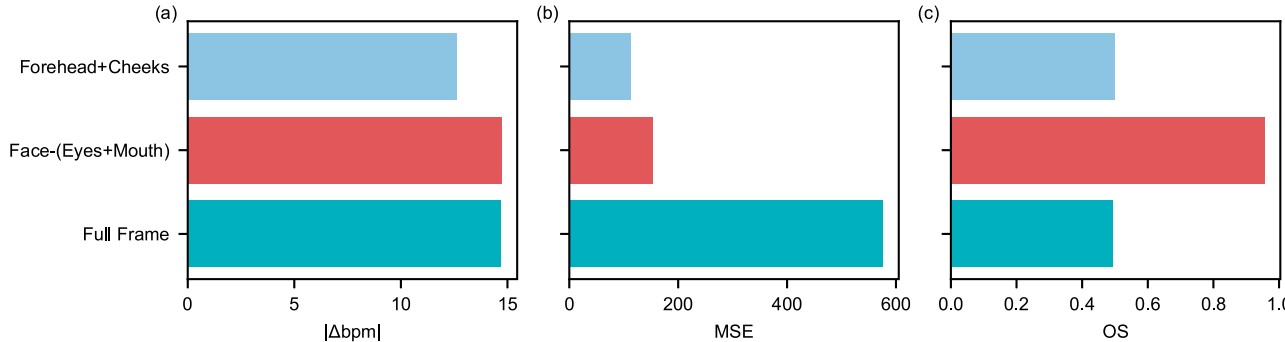

**Fig. 7 | Comparison of three regions of interest (ROIs) for frame modification.** This figure illustrates the evaluation of three ROIs in terms of: (**a**) average |Δbpm|, (**b**) average mean squared error (MSE), and (**c**) Overall evaluation score (OS). The results are averaged for one the subjects (*n*=1) in the LGI-PPGI dataset, for all the rPPG estimation techniques (*n*=5) and for all the activities (*n*=4) over the entire length of the videos. Source data is provided in Supplementary Table 5.

the least resilient, producing the highest bpm error. In addition, gym-related activities yielded the highest OS, but this was primarily due to the poor performance of the rPPG techniques. Talking activities also produced high OS values, which could be attributed to the natural lighting of the outdoor recording environment. The results of the study suggest that the facial ROI was the most effective of the three ROIs in balancing the produced bpm error and information loss, leading to the highest OS. However, from a computational and speed perspective, the full-frame ROI may be considered more optimal as it does not necessitate the extraction of facial mesh, leading to faster processing times.

The current investigation demonstrated that certain modification methods utilized in this study could elicit a commensurate bpm error while retaining substantial information within the frames. As a result, this study challenges conventional thinking and highlights that simpler techniques can achieve outcomes on par with other studies. Furthermore, these methods offer the advantage of not requiring rPPG signal extraction, allowing for real-time processing. In addition, most of these methods require few computational resources, making them feasible for use on low-computation devices, such as smartphones and surveillance cameras, or with online video conferencing applications. The findings of this study have implications beyond the field of physiological signal processing. Specifically, many of the modification methods tested herein share similarities with the filters commonly found on popular social media applications, including Instagram and Snapchat. These filters are used to add various effects to images and videos and are often adopted by users; according to the results of this study, they can indeed effectively conceal physiological signals as well.

## Limitations

This study adopted a balanced approach to maximize bpm errors and minimize data loss within the frame. However, the importance of these two factors may vary depending on the specific study objectives. Researchers should consider the unique goals of their study and determine the optimal balance between bpm errors and data loss accordingly. It is also important to note that the LGI-PPGI dataset used in this study has some limitations, particularly because it comprises primarily Caucasian subjects, and the videos are recorded in controlled settings, which may not reflect the diversity of real-world scenarios in which lighting conditions and camera angles vary considerably. In addition, the age range of the participants is limited, which may impact the generalizability of the results to a broader population.

It is essential to acknowledge that certain modification methods may yield slightly varied results upon reproduction. Methods involving the addition of noises inherently possess an element of randomness and may exhibit minor deviations with each execution. Similarly, time-averaging techniques are influenced by the frame rate, potentially leading to variations in results when applied to videos with differing frame rates.

## Guidance for future research

In the development of the modification pipeline, several challenges arose, shaping our approach. One notable challenge encountered was the selection of appropriate evaluation metrics. Our aim was to assess the algorithm's efficacy in concealing the rPPG signal, necessitating the analysis of various physiological parameters such as heart rate, blood pressure, and blood oxygen levels derived from the rPPG signal. Extensive review of existing literature revealed that heart rate measurement is the most well-established compared to other physiological parameters. Consequently, we prioritized the measurement of beats-per-minute differences as our primary metric for evaluating the modification pipeline. This decision was driven by the higher precision of algorithms in measuring heart rate compared to other parameters, thus allowing for more reliable assessment of algorithm performance.

Similarly, the selection of a suitable metric to quantify data loss between corresponding frames posed a challenge. We explored multiple metrics, including MSE and structural similarity index. While the structural similarity index offers a holistic measure of image similarity, we found that MSE provided finer granularity in distinguishing between different modification methods. Structural similarity index, on the other hand, often yielded similar scores for multiple methods, making it challenging to differentiate between them. As a result, we opted for MSE as it provided more discerning insights into the extent of data loss.

An interesting observation emerged during the development process regarding the impact of common image compression algorithms on concealing physiological signals. It became evident that inadvertent image compression could greatly influence the efficacy of modification methods in hiding the rPPG signal. To ensure the integrity of our results, we implemented rigorous measures within our software pipeline to prevent frame compression during video processing. This precaution was crucial in ensuring that the observed results were solely attributable to the modification methods themselves, independent of any extraneous factors such as video compression.

The recommendations derived from this study's findings are summarized below:

1. Further, develop modification methods: Researchers can continue to improve upon existing methods for modifying frames, potentially by exploring the combination of multiple simpler methods to produce better results. Furthermore, modification methods that can be reversed should also be explored.
2. Exploring Image Compression as Modification Methods: Our study revealed that image compression algorithms have the potential to assist in eliminating the rPPG signal. This presents an opportunity to investigate the effects of different lossy and lossless image compression techniques.
3. Explore Generative AI-Based Modification Pipelines: Recent advancements in generative artificial intelligence present an intriguing avenue

for research. Techniques such as diffusion have demonstrated remarkable capabilities in image-generation tasks. Exploring the application of diffusion-based models to inpaint facial regions to remove rPPG signals could yield promising results. Despite the typically large size of these models, strategies such as model distillation, quantization, and pruning offer potential avenues to optimize them for deployment on edge hardware.

4. Test using additional datasets: Although the LGI-PPG dataset is a widely used benchmark, it is important to test the algorithm against other datasets to evaluate its generalizability to different populations and settings.

5. Analyze the impact on other physiological parameters: It would be useful to investigate how the modification methods affect the performance of other physiological parameters beyond bpm, such as blood pressure, heart rate variability, and blood oxygen.

6. Investigate real-world scenarios: While the algorithm has been evaluated on pre-recorded facial videos, it is important to test its effectiveness in real-world scenarios, in which lighting and recording conditions may be more variable. To simulate practical conditions, existing datasets can also be augmented by adjusting parameters such as brightness, contrast, saturation, hue, and more. Additionally, incorporating transformations such as rotation, flipping, zooming in/out, and cropping can further emulate the appearance of images captured by security cameras, personal computers, and mobile phones. This approach can provide valuable insights into the algorithm's performance under realistic conditions.

## Conclusion

In summary, this study showcased the possibility of concealing physiological signals using simple frame modification methods while still maintaining frame quality to protect user privacy. The pipeline created in this study has enabled the use of different ROIs and the testing of 11 lightweight modification methods, including blurring, noise addition, and time-averaging techniques. Using five different rPPG techniques, the modified videos were evaluated according to the bpm errors they induced compared to the ground truth HR, and the results showed that the TA-S method is the most balanced in terms of preserving information within the frame and inducing the maximum bpm error, producing an average error of 22 bpm. The study also compared the speed of the algorithms and found them to require considerably fewer computational resources, enabling real-time implementation on low-compute devices. The facial ROI was also found to offer the best trade-off between bpm error and information retention. Overall, the study's findings contribute to the growing field of rPPG and the ongoing efforts to protect user privacy in videos without substantial distortion.

## Data availability

The LGI-PPGI dataset utilized in this study is publicly accessible and can be downloaded from the following website: [https://github.com/partofthestars/LGI-PPGI-DB]. All other data supporting the findings of this study is available within the paper and the Supplementary Information.

## Code availability

All code developed for the processing and analysis of the data is openly available under the MIT license without any constraints. It can be accessed from the code repository[26]. The applied rPPG methods are part of the pyVHR toolbox from https://github.com/phuselab/pyVHR.

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

## Acknowledgements
Saksham Bhutani is grateful for the support from ETH Zürich and Khalifa University (grant number RC2-2018-022).

## Author contributions
M.E. designed and led the study. S.B., M.E., and C.M. contributed to writing the original draft, investigation, methodology, visualization, and editing. All authors have read and agreed to the published version of the manuscript.

## Funding

## Competing interests
The authors declare no competing interests.
