## [Transparent Peer Review file · Communications Engineering]

Preserving Privacy and Video Quality through Remote Physiological Signal Removal

Corresponding Author: Professor Mohamed Elgendi

Version 0:

Reviewer comments:

Reviewer #2

(Remarks to the Author)

In this groundbreaking exploration of privacy-preserving techniques in rPPG, the authors present a study that addresses the concerns surrounding user privacy in the era of smart devices. The abstract introduces a rPPG method, emphasizing its accuracy in estimating physiological parameters while highlighting the ethical implications of continuous and surreptitious data collection. The researchers meticulously evaluate eleven ROI modifications, ranging from blurring to time-averaging techniques, across various activities. This study's strength lies in its commitment to real-world applicability, showcasing the Time-Averaging Sliding (TA-S) method as a standout performer, inducing a mere 22 beats per minute (bpm) error while preserving vital information within the frame. The conclusion emphasizes the study's contribution to the growing field of rPPG, suggesting its practical implications for protecting user privacy without compromising video quality on low-compute devices.

In conclusion, the paper's findings offer valuable insights into the delicate balance between concealing physiological signals and maintaining image quality, aligning with the ongoing efforts to address privacy concerns associated with rPPG technology. The researchers' meticulous approach and focus on practical applicability underscore the significance of their work in advancing the field. This pioneering study not only contributes to the evolving landscape of user privacy but also serves as an inspiring resource for researchers and practitioners engaged in the development and implementation of intelligent devices.

Reviewer #3

(Remarks to the Author)

This very well-written manuscript explores 11 different facial imaging frame modification methods to identify the caliber of quality each provides to conceal physiological signals in a computationally efficient fashion. The uniqueness of the work of interest to readership is the detailed development of the baseline dataset and the application of the different algorithms for image modification. The paper appears to establish that meaningful results can be accomplished with computationally efficient algorithms. Indeed, having this practical "review" of algorithms for the dataset indicated could serve as a baseline for other researchers to expand on possible new datasets. Table 3 and Figures 5 and 6 seem to be central outcomes that are useful to compare and contrast against the runtime evaluation of Figure 4.

The authors are candid about the limitations of the dataset, but this is secondary to the effort to make a first pass at how the routines summarized in Table 1 performed. Overall, the manuscript requires a very careful read to tease out the nuances of the methodology, but this need to be focused does not take away from the completeness of the research effort. Indeed, the authors have navigated quite a few steps in the development of the baseline data set and application of algorithms that must have reflected many challenges not described in the manuscript. From that perspective it was hoped that the discussion would have been more granular on challenges experienced in application effort to achieve the results, and what considerations were made (if any) to move forward when computational challenges arose. This would help those wishing to carry on with the work some sense of decisions (if any) that, for instance, may want to act on the recommendation to "further develop modification methods" or to analyze the impact of other physiological parameters. Were any initial forays into the recommended "next steps" made but initially non-conclusive and dropped for any reason? Further, how might one prioritize the Discussion points if the work were to be reproduced by others? What would you do differently to process the basic dataset, if anything?

Overall, this is an informative manuscript that outlines some initially encouraging work in the important area of image de-identification that has potential to inspire other researchers to undertake similar research.

Version 1:

Reviewer comments:

Reviewer #3

(Remarks to the Author)

Thank you for the thoroughness of your responses to the review comments on your manuscript. Your careful attention to each point has effectively addressed all concerns raised during the review process. The manuscript now is considered acceptable as well as quite impressive. Thank you for your diligence and dedication throughout this process.

Reviewer 1:

1. In this groundbreaking exploration of privacy-preserving techniques in rPPG, the authors present a study that addresses the concerns surrounding user privacy in the era of smart devices. The abstract introduces a rPPG method, emphasizing its accuracy in estimating physiological parameters while highlighting the ethical implications of continuous and surreptitious data collection. The researchers meticulously evaluate eleven ROI modifications, ranging from blurring to time-averaging techniques, across various activities. This study's strength lies in its commitment to real-world applicability, showcasing the Time-Averaging Sliding (TA-S) method as a standout performer, inducing a mere 22 beats per minute (bpm) error while preserving vital information within the frame. The conclusion emphasizes the study's contribution to the growing field of rPPG, suggesting its practical implications for protecting user privacy without compromising video quality on low-compute devices.

In conclusion, the paper's findings offer valuable insights into the delicate balance between concealing physiological signals and maintaining image quality, aligning with the ongoing efforts to address privacy concerns associated with rPPG technology. The researchers' meticulous approach and focus on practical applicability underscore the significance of their work in advancing the field. This pioneering study not only contributes to the evolving landscape of user privacy but also serves as an inspiring resource for researchers and practitioners engaged in the development and implementation of intelligent devices.

Author reply: We sincerely thank you for your encouraging and detailed feedback on our manuscript. Your positive remarks and the acknowledgment of our efforts is greatly appreciated. Your summary not only captures the essence of our study but also reinforces our commitment to advancing privacy-preserving technologies.

Reviewer 2:

1. This very well-written manuscript explores 11 different facial imaging frame modification methods to identify the caliber of quality each provides to conceal physiological signals in a computationally efficient fashion. The uniqueness of the work of interest to readership is the detailed development of the baseline dataset and the application of the different algorithms for image modification. The paper appears to establish that meaningful results can be accomplished with computationally efficient algorithms. Indeed, having this practical "review" of algorithms for the dataset indicated could serve as a baseline for other researchers to expand on possible new datasets. Table 3 and Figures 5 and 6 seem to be central outcomes that are useful to compare and contrast against the runtime evaluation of Figure 4. The authors are candid about the limitations of the dataset, but this is secondary to the effort to make a first pass at how the routines summarized in Table 1 performed. Overall, the manuscript requires a very careful read to tease out the nuances of the methodology, but this need to be focused does not take away from the completeness of the research effort.

Author reply: Thank you for your insightful feedback and recognition of our manuscript's contribution to the field. We appreciate your acknowledgment of its clarity, the detailed development of the baseline dataset, and the practical application of image modification algorithms. Your comments reinforce the value of our research and its potential as a baseline for future studies.

2. Indeed, the authors have navigated quite a few steps in the development of the baseline data set and application of algorithms that must have reflected many challenges not described in the manuscript. From that perspective it was hoped that the discussion would have been more granular on challenges experienced in application effort to achieve the results, and what considerations were made (if any) to move forward when computational challenges arose. This would help those wishing to carry on with the work some sense of decisions (if any) that, for instance, may want to act on the recommendation to “further develop modification methods” or to analyze the impact of other physiological parameters.

Author reply: Thank you for your constructive critique. Recognizing the importance of detailing the challenges and decision-making processes in our work, we've revised the discussion section with specifics on the hurdles encountered and the strategic choices made during the development of our modification techniques. This update aims to provide a clearer roadmap for researchers interested in advancing this field.

Author action 1: We have introduced a new subsection, "Guidance for Future Research," to elaborate on the obstacles we faced and the rationale behind our methodological choices. The added content is as follows:

“In the development of the modification pipeline, several challenges arose, shaping our approach. One significant challenge encountered was the selection of appropriate evaluation metrics. Our aim was to assess the algorithm's efficacy in concealing the rPPG signal, necessitating the analysis of various physiological parameters such as heart rate, blood pressure, and blood oxygen levels derived from the rPPG signal. Extensive review of existing literature revealed that heart rate measurement is the most well-established compared to other physiological parameters. Consequently, we prioritized the measurement of beats-per-minute differences as our primary metric for evaluating the modification pipeline. This decision was driven by the higher precision of algorithms in measuring heart rate compared to other parameters, thus allowing for more reliable assessment of algorithm performance.

Similarly, the selection of a suitable metric to quantify data loss between corresponding frames posed a challenge. We explored multiple metrics, including MSE and structural similarity index (SSIM). While SSIM offers a holistic measure of image similarity, we found that MSE provided finer granularity in distinguishing between different modification methods. SSIM, on the other hand, often yielded similar scores for multiple methods, making it challenging to differentiate between them. As a result, we opted for MSE as it provided more discerning insights into the extent of data loss.”

Author action 2: We have included our observation on potential pitfalls that future researchers might encounter, as detailed below:

“An interesting observation emerged during the development process regarding the impact of common image compression algorithms on concealing physiological signals. It became evident that inadvertent image compression could significantly influence the efficacy of modification methods in hiding the rPPG signal. To ensure the integrity of our results, we implemented rigorous measures within our software pipeline to prevent frame compression during video processing. This precaution was crucial in ensuring that the observed results were solely attributable to the modification methods themselves, independent of any extraneous factors such as video compression.”

3. Were any initial forays into the recommended “next steps” made but initially non-conclusive and dropped for any reason?

Author reply: Thank you for your question. Indeed, certain recommendations for future research were initially explored but did not yield conclusive results at the outset, leading to their omission in the initial submission. Recognizing the value of your question, we have now included these exploratory ideas in the manuscript to provide a more comprehensive perspective.

Author action 1: We've incorporated a point in the recommendations on the potential use of generative models, such as diffusion-based techniques, for effectively removing rPPG signals from facial videos:

“Explore Generative AI-Based Modification Pipelines: Recent advancements in generative artificial intelligence (AI) present an intriguing avenue for research. Techniques such as diffusion have demonstrated remarkable capabilities in image generation tasks. Exploring the application of diffusion-based models to inpaint facial

regions to remove rPPG signals could yield promising results. Despite the typically large size of these models, strategies such as model distillation, quantization, and pruning offer potential avenues to optimize them for deployment on edge hardware.”

Author action 2: We have added another point based on our observations on how various image compression algorithms could aid in the removal of physiological signals, underscoring their potential role in privacy preservation:

“Exploring Image Compression as Modification Methods: Our study revealed that image compression algorithms have the potential to assist in eliminating the rPPG signal. This presents an opportunity to investigate the effects of different lossy and lossless image compression techniques.”

4. Further, how might one prioritize the Discussion points if the work were to be reproduced by others?

Author reply: Thank you for highlighting this. Reproducibility and extendibility of this work is paramount to us. To facilitate this, we have made available a comprehensive code repository that includes everything needed to replicate our study's findings and extend upon them.

Author action 1: We have created a public code repository that features an exhaustive readme detailing the setup requirements, installation instructions, file structure and functionality, usage examples, and dataset information. A thorough demonstration notebook within this repository guides users through applying the modification pipeline with various filters and ROIs, evaluating information loss and runtime efficiency, and developing new filters and metrics. To inform readers, we have added a code availability statement in our manuscript, directing them to the repository at <https://github.com/saksham2001/rPPG-removal> for full access to the resources necessary for replication and further research.

Author action 2: To enhance the manuscript's utility for future research, we introduced a "Considerations for reproduction" subsection within the discussions. This new section provides comprehensive details about the publicly available code repository designed to support the reproducibility and extendibility of our work. Highlighted in this segment is the pyRemoval package, a core component of our research, which facilitates video processing and the application of modification methods. The section underscores the package's adaptability to various datasets beyond the LGI-PPGI dataset used in our study, its tested compatibility across different platforms, and includes a detailed demonstration notebook offering step-by-step instructions for employing the package, developing new video modification filters, integrating novel ROIs, and employing alternative metrics for analyzing information loss and runtime speed. Furthermore, this subsection draws attention to potential variations in some modification methods due to the inherent randomness or dependence upon some parameters.

“The source code employed in this study has been made publicly accessible to promote reproducibility and extendibility. Specifically, the pyRemoval package within the source code encompasses all the tools necessary for video processing, including the modification methods discussed in this study. Additionally, the package offers the functionality to analyse the processed videos, including the assessment of information loss and runtime speed. Importantly, the source code is not restricted to the LGI-PPGI dataset and is adaptable to other datasets. Thorough installation instructions are provided alongside the source code, ensuring straightforward deployment. Extensive testing has been conducted across different platforms to verify the compatibility. A detailed demonstration notebook is included, with comprehensive guidance on the utilization of the package.

Our aim is to establish this codebase as a foundational resource for researchers to build upon. To facilitate this, we have provided boilerplate code along with detailed instructions for creating new filters for video modification, integrating new ROIs, and analyzing methods using alternative metrics to quantify information loss and runtime speed.

It is essential to acknowledge that certain modification methods may yield slightly varied results upon reproduction. Methods involving the addition of noises inherently possess an element of randomness and may exhibit minor deviations with each execution. Similarly, time-averaging techniques are influenced by the frame rate, potentially leading to variations in results when applied to videos with differing frame rates.”

5. What would you do differently to process the basic dataset, if anything/?

Author reply: Thank you for raising this important question. As we've acknowledged in the limitations section, the datasets currently available in this domain face significant constraints in terms of subject diversity and recording settings, which may not adequately represent the variety found in real-world environments. To enhance the dataset's diversity and improve the generalizability of our results, we could have implemented several adjustments. Specifically, augmenting the dataset by modifying parameters such as brightness, contrast, saturation, and hue could make it more reflective of real-life scenarios. Furthermore, applying transformations like rotation, flipping, zooming in/out, and cropping would help mimic the conditions under which images are captured by different devices, such as security cameras, personal computers, and mobile phones.

Author action: In addition to acknowledging these points in the limitations subsection, we have refined our recommendations for future research to provide some ideas to researchers working on this topic:

“Investigate real-world scenarios: While the algorithm has been evaluated on pre-recorded facial videos, it is important to test its effectiveness in real-world scenarios, in which lighting and recording conditions may be more variable. To simulate practical conditions, existing datasets can also be augmented by adjusting parameters such as brightness, contrast, saturation, hue, and more. Additionally, incorporating transformations such as rotation, flipping, zooming in/out, and cropping can further emulate the appearance of images captured by security cameras, personal computers, and mobile phones. This approach can provide valuable insights into the algorithm's performance under realistic conditions.”

6. Overall, this is an informative manuscript that outlines some initially encouraging work in the important area of image de-identification that has potential to inspire other researchers to undertake similar research.

Author reply: Thank you for recognizing the significance and potential impact of our work in the realm of image de-identification. We share your belief that as remote physiological measurement technologies advance, the need to effectively remove these signals from visual modalities will become increasingly critical. Our study represents an initial step towards addressing this challenge, and we are optimistic that it will inspire further research aimed at developing robust algorithms for the removal of physiological data from images.

Reviewer 3:

1. Thank you for the thoroughness of your responses to the review comments on your manuscript. Your careful attention to each point has effectively addressed all concerns raised during the review process. The manuscript now is considered acceptable as well as quite impressive. Thank you for your diligence and dedication throughout this process.

Author reply: Thank you for your kind remarks. Your thorough reviews have been invaluable in helping us significantly improve the quality and value of this article. We are grateful for your diligence and thoughtful feedback throughout the review process.